# Dominant ionic currents in rabbit ventricular action potential dynamics

**Zhechao Yang[1], Hao Gao[1], Godfrey L. Smith[2], Radostin D. Simitev[1]***

**1** School of Mathematics and Statistics, University of Glasgow, Glasgow, United Kingdom, **2** School of Cardiovascular and Metabolic Health, University of Glasgow, Glasgow, United Kingdom

* Radostin.Simitev@glasgow.ac.uk

**Data availability statement:** Data and code are available from https://github.com/Zhechao Yanggla/Sobol_paper.

## Abstract

Mathematical models of cardiac cell electrical activity include numerous parameters, making calibration to experimental data and individual-specific modeling challenging. This study applies Sobol sensitivity analysis, a global variance-decomposition method, to identify the most influential parameters in the Shannon model of rabbit ventricular myocyte action potential (AP). The analysis highlights the background chloride current ($I_{Clb}$) as the dominant determinant of AP variability. Additionally, the inward rectifier potassium current ($I_{K1}$), fast/slow delayed rectifier potassium currents ($I_{Kr}$, $I_{Ks}$), sodium-calcium exchanger current ($I_{NaCa}$), the slow component of the transient outward potassium current ($I_{tos}$), and L-type calcium current ($I_{CaL}$) significantly affect AP biomarkers, including duration, plateau potential, and resting potential. Exploiting these results, a hierarchical reduction of the model is performed and demonstrates that retaining only six key parameters can capture sufficiently well individual biomarkers, with a coefficient of determination exceeding 0.9 for selected cases. These findings improve the utility of the Shannon model for personalized simulations, aiding applications like digital twins and drug response predictions in biomedical research.

## Author summary

In our study, we explored the complexity of a mathematical model used to understand the electrical activity of rabbit ventricular cells. Such models involve many parameters, which makes it difficult to match them with experimental data or to personalize them for individual cells. To address this, we used a method for global sensitivity analysis to identify which ion current parameters have the most impact on the action potential output of the model. Our analysis shows that the background chloride current is the main factor influencing the variability of the rabbit ventricular action potential. Other important currents include the inward rectifier potassium current, various potassium currents, sodium-calcium exchanger current, the slow component of the transient outward potassium current, and the L-type calcium current. These factors play a significant role in determining key features of the action potential, such as its duration, plateau potential,

**Funding:** This work was supported by the UK Engineering and Physical Sciences Research Council [grant numbers EP/S030875/1 and EP/T017899/1]. The funders had no role in study design, data collection and analysis, decision to publish, or preparation of the manuscript.

**Competing interests:** The authors have declared that no competing interests exist.

and resting potential. Based on these results, we proposed a sequence of simplified models obtained by retaining variability in a few most influential parameters while fixing the rest to appropriate constant values. This approach is also applicable to other mathematical models of cardiac action potentials and can make them more useful for personalized simulations, and in areas like digital twins and predicting drug responses in biomedical research.

## Introduction

Mathematical models of cardiac action potentials (APs) are essential tools for studying the electrical activity of cardiac cells and understanding their physiological and pathological behaviors. In this work, we conduct a global sensitivity analysis of a mathematical model of rabbit ventricular myocyte APs, and rank key parameters based on their impact on aspects of the model's output.

Ventricular APs are transient deviations from the resting electrical potential across the sarcolemma, and play an essential role in cardiac function. They are triggered by electrical stimuli and dynamically sustained by ionic conductance changes regulated by voltage and ion concentrations. The complexity of the underlying cellular processes and structures necessitates nonlinear, stiff, and high dimensional mathematical models of the AP [1]. Consequently, their analysis, numerical simulation, and comparison with experimental data is challenging. Often a model reduction is needed to a simplified system while retaining the predictive power of the original model.

A widely used approach for model reduction and simplification is sensitivity analysis. This method can be viewed as an optimization problem, where the objective is to minimize the model's dimensionality while ensuring that the discrepancy between the outputs of the original and reduced models remains sufficiently small. In practice, this involves identifying model quantities whose variations have negligible impact on the output and replacing them with appropriate constants. Compared to other model reduction techniques such as variable lumping, coordinate transformations, manifold reduction, truncation via singular value decomposition, or homogenization, sensitivity analysis offers a key advantage: it retains the physiological interpretability of model variables, parameters, and components [2].

This study is motivated by the need to model recent experimental measurements reported by [3]. In that work, AP waveforms were measured in hundreds of isolated rabbit ventricular myocytes, both before and after drug administration. Cells were sourced from various regions of the left ventricle across multiple animals. The most surprising finding was that AP waveforms varied more significantly between individual cells than between regions or even between different animals. Cell-specific mathematical models are required to capture this pronounced cell-to-cell variability. Such models can be developed by adjusting a subset of parameters in a generic baseline model, such as [4]. To achieve this, a critical step is understanding how variations in parameter values influence the AP waveform. The overall goal of this study is to perform this sensitivity analysis, with specific objectives outlined further below.

There is a substantial body of literature on sensitivity analysis of detailed cardiac AP models, with a selection of the most notable studies summarized in Table 1. Among these, only the work of [16] specifically addressed rabbit ventricular AP models. Their study systematically investigated how parameter variations affect a wide range of electrophysiological properties, including steady-state AP and intracellular calcium concentrations ($[Ca^{2+}]_i$), AP duration and triangulation, AP duration rate dependence and restitution curves, and AP duration adaptation to abrupt changes in basic cycle length (BCL). A major limitation of [16] is its

**Table 1. Selected works on sensitivity analysis of cardiac cellular electrophysiology models.**

| Goals | Methods & Models | Results |
|---|---|---|
| Gemmell et al. [5] | | |
| • To perform a systematic exploration of the effects of simultaneously varying the magnitude of six transmembrane current conductances in two rabbit-specific ventricular AP models. | • Clutter-based dimension reordering. • Rabbit-specific ventricular AP model: [4]. | • $g_{Ca,L}$ had the greatest influence on AP duration variability at both 400 and 1000 ms, along with $g_{K1}$ and $g_{to}$ at 400 and 1000 ms, respectively. |
| Sobie [6] | | |
| • To investigate how maximum ion channel conductances affect the outputs of a simplified cardiac cell model, specifically focusing on AP shape and restitution. | • Multivariable Regression. • Cardiac cell models: [7–9]. | • Blocking rapid delayed rectifier current ($I_{Kr}$) causes a greater prolongation of AP duration than blocking slow delayed rectifier current ($I_{Ks}$), which has little effect. Changes in inward rectifier current ($I_{K1}$) have a much greater effect on AP duration than changes in $I_{Kr}$ in the model of [8]. |
| Coveney & Clayton [10] | | |
| • To study how different ionic conductances in two cardiac cell models affect the AP duration and other electrophysiological properties. | • Variance based sensitivity analysis. • Cardiac cell models: [11,12]. | • Changes in maximum conductance of the ultra-rapid K + channel ($I_{Kur}$) would have opposite effects on AP duration. |
| Romero et al. [13] | | |
| • To investigate the impact of variability in ionic currents on the electrophysiological properties of human ventricular cells. | • Local sensitivity analysis. • Human ventricular model: [14]. | • AP duration is moderately sensitive to changes in maximal conductances of all currents involved in repolarization and also $I_{Ks}$ and $I_{CaL}$ kinetics. |
| Chang et al. [15] | | |
| • To investigate how uncertainty in input parameters influences electrophysiological features and mechanical variables. | • Variance based sensitivity analysis. • Cardiac cell model: [7]. | • The analysis showed that parameters like $G_{si}$, $G_K$, and $G_b$ were crucial in determining the variability in AP duration, a key electrophysiological measure. |
| Romero et al. [16] | | |
| • To characterise the sensitivity of selected cellular biomarkers of arrhythmic risk to ionic current properties in rabbit ventricular models. • To compare with experimental data. | • Local sensitivity analysis.• Rabbit cell models: [4,17]. | • AP duration is significantly influenced by most repolarization currents. • APtriangulation is mainly regulated by $I_{K1}$.• AP duration restitution properties, and $Ca_i^{2+}$ rate dependence arestrongly affected by $I_{NaK}$, $I_{CaL}$ and $I_{NaCa}$. |
| Johnstone et al. [18] | | |
| • To investigate the impact of variability in ionic currents on the electrophysiological properties of human ventricular cells. | • Variance based sensitivity analysis. • Human ventricular model: [14]. | • $V_{max}$ is sensitive to $G_{Na}$, $G_{CaL}$. $V_m$, $A_{90}$ and $A_{50}$ are both influenced by a similar group of inputs, and resting voltage is influenced by $G_{K1}$ and pump currents. |
| Del Corso et al. [19] | | |
| • To identify which of the uncertain inputs mostly affect electrophysiology of the left ventricle. | • Variance based sensitivity analysis. • Cardiac cell model: [14]. | • The most influential input parameters for AP duration and shape similarity are the maximal IKr, IKs and ICal conductances ($G_{Kr}$, $G_{Ks}$ and $G_{CaL}$) along with the extracellular $Ca$ and $Na$ concentration. |
| Chang & Clayton [20] | | |
| • To investigate the impact of variability in ionic currents on the electrophysiological properties of human ventricular cells. | • Variance based sensitivity analysis. • Cardiac cell model: [11]. | • AP upstroke (max dV/dt) and maximum voltage were highly sensitive to $G_{Na}$. Dome voltage was most sensitive to $G_{Kur}$ and $G_{CaL}$, whilst $G_{K1}$ had a strong effect on $A_{90}$ and resting voltage. $A_{50}$ was most sensitive to $G_{CaL}$. |

(*Continued*)

**Table 1**. (Continued)

| Goals | Methods & Models | Results |
|---|---|---|
| Pathmanathan et al. [21] | | |
| • To demonstrate feasibility of performing comprehensive UQ/SA for cardiac cell models and demonstrate how to assess robustness and overcome model failure when performing cardiac UQ analyses. | • Variance based sensitivity analysis. • Human ventricular model: [14]. | • $E_h$, plays a dominant role in determining MaxUpstrokeVelocity. TimeOfMaxUpstrokeVelocity is controlled by three parameters $E_m$, $k_m$ and $E_z$. ($E_h$, $\log(\delta_h)$, $E_r$, $E_d$, $E_f$), which were the parameters identified to be highly influential. |
| Mora et al. [22] | | |
| • To study the electrical activity and ionic homeostasis of failing myocytes. • To compare with experimental data. | • Single-parameter sensitivity analysis. • Human ventricular model: modified [23]. | • $I_{NaL}$ and $I_{NaK}$ are the most important contributors to AP duration variations. SERCA plays an important role in modulating $Ca^{2+}$, with the $Na^{2+}/Ca^{2+}$ exchanger (NCX) and other $Ca^{2+}$ cycling proteins also playing a significant role. |
| Sadrieh et al. [24] | | |
| • To extend existing sensitivity analyses to electrocardiogram (ECG) signals derived from multicellular systems and quantify the contribution of ionic conductances to emergent properties of the ECG. | • Partial least squares analysis. • Human ventricular model: [23]. | • $A_{90}$ is approximately fourfold more sensitive to changes in $G_{CaL}$ than in isolated cells but less sensitive to changes in $G_{K1}$ and $G_{ncx}$. |
| Parikh et al. [25] | | |
| • To apply global sensitivity analysis to the existing Comprehensive in vitro Proarrhythmia Assay (CiPA) in silico framework to identify the key model components that derived metrics are most sensitive to. | • Variance based sensitivity analysis. • Human ventricular model: CiPAORd model, modified from [23]. | • The Sobol sensitivity indices indicate that $A_{90}$ is the most sensitive to sbIKr block, qNet to sbINaL, and peakCa to bICaL. |

use of local sensitivity analysis, where parameters are varied by fixed percentages ($\pm 15\%$ and $\pm 30\%$) around Shannon model baseline values. This approach is valid only near reference values and ignores interactions between multiple parameters. Experimental data from [3] show significant variability that the baseline Shannon model cannot capture, and its nonlinearity suggests parameter sensitivity varies across the space. Thus, global sensitivity analysis, which evaluates parameter influence across the full space and considers nonlinear interactions, is essential for a more complete understanding of model behaviour.

Several methods for global sensitivity analysis, including partial rank correlations and Fourier amplitude sensitivity, are well-documented in the literature [26]. For this study, we have chosen the Sobol global sensitivity method [27], based on its robustness and efficiency. Sobol's method quantifies the contribution of input variables or parameters to the variance of a model's output, helping to identify the most influential factors. A brief informal outline of the theoretical concepts behind Sobol analysis is provided in further below.

Our work has three main objectives. First, we estimate Sobol sensitivity indices for all maximal current strength parameters in the Shannon model, focusing on their influence on selected AP biomarkers, including duration, plateau potential, and resting potential. Second, we rank these parameters based on their grand total Sobol sensitivity indices to identify the most influential ionic currents. Third, we validate these rankings and propose a hierarchy of reduced models, demonstrating that dimensionality reduction is feasible without significant loss of predictive accuracy.

## Methods and models

### Shannon's AP model and output biomarkers

To understand the cell-to-cell variability reported in [3], we consider the [4] mathematical model for the rabbit ventricular myocyte AP. This model, along with that of [17], is among the two widely used models for this cell type. We chose the Shannon model as it better captures the measurements of [3], as detailed in their analysis. As typical for models of myocyte electrophysiology, the Shannon model takes the form of a system of ordinary differential equations

$$
\frac{d}{dt}V(t) = -\left(\sum_{i=1}^{N} p_i I_i(V, \mathbf{z}, \mathbf{c}, \boldsymbol{\theta}) + I_{\text{stim}}(t, \mathbf{u})\right),
$$

$$
\frac{d}{dt}\mathbf{z}(t) = \mathbf{g}(V, \mathbf{z}, \boldsymbol{\theta}), \tag{1}
$$

$$
\frac{d}{dt}c_k(t) = s_k\left(\sum_{j_k} p_{j_k} I_{j_k}(V, \mathbf{z}, \mathbf{c}, \boldsymbol{\theta})\right), \quad j_k \subseteq \{x \in \mathbb{N} \mid 1 \leq x \leq N\}.
$$

Here, $V$ is the electric potential across the cell membrane, $t$ represents time. Vector $\mathbf{z}$ represents variables describing channel gating configurations such as activation and inactivation or Markov model states, and vector $\mathbf{c}$, with components $c_k$, represents ionic intracellular concentrations, while $\mathbf{g}$ and $s_k$ are functional dependences. Model parameters, such as maximal conductances and channel kinetics, are contained in the vector $\boldsymbol{\theta}$, while $\mathbf{u}$ represents external protocol parameters, including stimulus timing, duration, and strength. The transient changes in membrane potential, known as APs, are controlled by the sum of currents $I_i$ flowing across the membrane or between internal compartments (e.g., organelles). The Shannon model includes $N = 15$ distinct currents, detailed in Table 2. Their explicit formulations and parameter values are available in [4]. The factors $p_i$ multiply the ionic currents everywhere the latter appear, and represent increase or decrease of current strengths relative to their baseline values. They have been embedded in the model for the purposes of the sensitivity analysis, as they allow the original model formulation to be used without modification. A stimulus current, $I_{\text{stim}}$, is used to excite the cell and is applied at a basic cycle length ($t_{\text{BCL}}$). Under physiological pacing conditions, the system reaches a periodic train of APs irrespective of the initial conditions.

The model equations are solved numerically. To avoid coding errors, we use a machine readable model specification file available from the CellML model repository [28]. Numerical integration is done using the Myokit suite for cardiac cellular electrophysiology simulations [29], which in turn employs solvers for nonlinear differential/algebraic equation from the SUNDIALS library [30].

Due to the large number of parameters in the Shannon model, it is not computationally feasible to include all in a sensitivity analysis. Following previous studies e.g. [3,16], we restrict the attention to investigating the influence of the relative strengths of ionic currents. To measure these we introduce $N = 15$ auxiliary parameters denoted by $p_i$ in Eq (1). These can be interpreted as scaling factors of the maximal conductances for currents of the Ohmic form, or of the max $I_i$ for currents of the Goldman-Hodgkin-Katz form. In essence, the factors $p_i$ represent the relative strengths of ionic currents compared to their "baseline" values published in [4].

The influence of the sensitivity parameters $p_i$ on the model will be assessed by measuring the variation of $K = 6$ biomarkers $y_i$ that capture important characteristics of the AP waveform $V(t; p_1, p_2, \ldots, p_N)$. These model output quantities are listed and defined in Table 3. In

**Table 2. The fifteen ionic currents of the Shannon mathematical model for the rabbit ventricular myocyte AP [4].**

| Ionic current, $I_i$ | Description |
|---|---|
| $I_{CaL}$ | L-type $Ca^{2+}$ current. |
| $I_{Cab}$ | Background $Ca^{2+}$ current. |
| $I_{Cap}$ | $Ca^{2+}$ pump current. |
| $I_{ClCa}$ | $Ca^{2+}$-activated $Cl^-$ current. |
| $I_{K1}$ | Inward rectifier $K^+$ current / Time-independent $K^+$ current. |
| $I_{Kp}$ | Background potassium $K^+$ current. |
| $I_{Kr}$ | Fast delayed rectifier $K^+$ current. |
| $I_{Ks}$ | Slow delayed rectifier $K^+$ current. |
| $I_{Clb}$ | $Cl^-$ background current. |
| $I_{Na}$ | Fast Na current. |
| $I_{NaCa}$ | $Na^+/Ca^{2+}$ exchanger current. |
| $I_{NaK}$ | Na-K pump current. |
| $I_{Nab}$ | Background $Na^+$ current. |
| $I_{tof}$ | Fast component of the transient outward potassium current. |
| $I_{tos}$ | Slow component of the transient outward potassium current. |

**Table 3. A list of the six AP output biomarkers $y^k$ used for sensitivity analysis, and definitions of their corresponding relationships to the parameters of the Shannon model, $y^k = f^k(p_1, p_2, \ldots, p_N)$. The dependence of $y^k$ on parameters is due to the voltage potential being parameter dependent, $V = V(t; p_1, p_2, \ldots, p_N)$, as discussed in the text.**

| AP biomarker | Symbol $y^k$ | Definition (Description) |
|---|---|---|
| Peak potential | $V_{max}$ | $V_{max} = \max_t V(t)$. |
| Resting potential (alt. maximal diastolic potential) | $V_{rest}$ | $V_{rest} = V(t_{BCL})$, where $t_{BCL}$ is the "basic cycle length". |
| AP duration at 90% repolarization | $A_{90}$ | $A_{90} = V^{-1}\left(V_{max} - 0.9\lvert V_{max} - V_{rest}\rvert\right) - t_0$ (The time from stimulus to the moment where the potential reaches 90% of full repolarization as defined by the difference between peak and resting potential.) |
| AP duration at 30% repol. | $A_{30}$ | Ditto. |
| Plateau potential | $V_{plt}$ | $V_{plt} = V(t_0 + A_{90}/2)$. |
| Bulk | $B$ | $B = \int_0^{t_{BCL}} \lvert V(t) - V_{rest}\rvert \, dt$. |

the Table and throughout the rest of the text $t_0$ and $t_{BCL}$ refer to the start and the end of the last AP, and $V(t)$ refers to its voltage trace. In summary, the relationships between biomarkers and parameters can be represented formally by a mapping $\mathbf{f}$ as

$$\mathbf{y} = \mathbf{f}(\mathbf{p}), \quad \mathbf{p} = (p_1, p_2, \ldots, p_N). \tag{2}$$

## Sobol's sensitivity analysis of variance decomposition

We employ a global method for sensitivity analysis based on variance decomposition, originally proposed by [31]. The starting point is the relationship (2) between AP output biomarkers $\mathbf{y}$ and the relative strengths of ionic currents $\mathbf{p}$. The sensitivity of each biomarker $y^k$ can be analyzed independently of the others. For simplicity, we drop the superscript $k$ and focus on a

single scalar model output, $y = f(\mathbf{p})$. To ensure that biomarker values are comparable in magnitude, the values are then normalized $\tilde{y} = \tilde{f}(\mathbf{p})$ by computing their standard z-score $\tilde{f}(\mathbf{p}) = z(f(\mathbf{p}))$, defined as $z(x_i) = (x_i - \mu)/\sigma$ for a general set of values $\{x_i, \ i = 1, \dots, L\}$ with mean and standard deviation $\mu$ and $\sigma$, respectively.

Consider the parameter vector as a vector of $N = 15$ independent random variables, $\mathbf{P} = (p_1, p_2, \dots, p_N)$. The standardized biomarker $\tilde{Y}$ is then also a random variable, given by $\tilde{Y} = \tilde{f}(\mathbf{P}) = f(p_1, p_2, \dots, p_N)$, where $f$ is a deterministic function specified in Table 3 and consequently so is $\tilde{f}$. It can be shown that any function of independent random variables has a functional decomposition of the form

$$\tilde{f}(\mathbf{P}) = \tilde{f}_0 + \sum_{i=1}^{N} \tilde{f}_i(p_i) + \sum_{1 \leq i < j \leq N} \tilde{f}_{ij}(p_i, p_j) + \dots + \tilde{f}_{1,2,\dots,N}(p_1, p_2, \dots, p_N), \tag{3a}$$

where the quantities $\tilde{f}_I(p_I)$ are defined as:

$$\tilde{f}_I(p_I) = \mathbb{E}[\tilde{f}(\mathbf{P}) \mid p_I] - \sum_{J \subset I} \tilde{f}_J(p_J), \tag{3b}$$

for any subset of indices $I \subseteq \{1, 2, \dots, N\}$, and $\mathbb{E}[\cdot]$ is the expectation value operator. These quantities represent the interaction "effects" on the output $\tilde{Y}$ of all variables indexed by the set $I$, where the sum is taken over all proper subsets $J$ of $I$. For example,

$$\tilde{f}_0 = \mathbb{E}[\tilde{f}(\mathbf{P})],$$
$$\tilde{f}_i(p_i) = \mathbb{E}[\tilde{f}(\mathbf{P}) \mid p_i] - \tilde{f}_0,$$
$$\tilde{f}_{ij}(p_i, p_j) = \mathbb{E}[\tilde{f}(\mathbf{P}) \mid p_i, p_j] - \tilde{f}_i(p_i) - \tilde{f}_j(p_j) - \tilde{f}_0,$$

represent the overall mean value of the output, the correction effects due to the isolated "action" of each single parameter (first-order effects), and the correction effects due to the interactions of all possible combinations of two parameters (second-order interaction effects), respectively. The decomposition (3a) can be easily verified by nested back-substituting (3b) into (3a), and taking into account the fact that the conditional expectation of a deterministic function in the case all of its values are deterministic is equal to the value of the function itself, formally

$$\mathbb{E}[\tilde{f}(p_1, p_2, \dots, p_N) \mid p_1 = p_1, p_2 = p_2, \dots, p_N = p_N] = \tilde{f}(p_1, p_2, \dots p_N).$$

The standard measure of the variation of the biomarker $\tilde{Y}$ is the variance $\mathbb{V}[\tilde{f}(\mathbf{P})]$. Substituting the functional decomposition (3) into the definition of variance:

$$\mathbb{V}[\tilde{f}(\mathbf{P})] = \mathbb{E}[(\tilde{f}(\mathbf{P}) - \mathbb{E}[\tilde{f}(\mathbf{P})])^2],$$

expanding the products, and using the fact that cross terms are orthogonal

$$\mathbb{E}[\tilde{f}_I(P_I)\tilde{f}_J(P_J)] = 0 \quad \text{for} \quad I \neq J, \quad I, J \subseteq \{1, 2, \dots, N\},$$

along with the property

$$\mathbb{V}[\tilde{f}_I(P_I)] = \mathbb{E}[\tilde{f}_I(P_I)^2],$$

we find that the total variance has the decomposition:

$$\mathbb{V}[\tilde{f}(\mathbf{P})] = \sum_{i=1}^{N} \mathbb{V}[\tilde{f}_i(p_i)] + \sum_{1 \leq i < j \leq N} \mathbb{V}[\tilde{f}_{ij}(p_i, p_j)] + \cdots + \mathbb{V}[\tilde{f}_{1,2,\dots,N}(p_1, p_2, \dots, p_N)]. \tag{4}$$

With this, direct measures of the "main effect" of a given parameter $p_i$, and the "higher-order effects" of interactions between combinations of parameters on the output $\tilde{Y}$ are defined as the corresponding "partial" variances in the decomposition (4) normalised by the total variance,

$$S_i = \frac{\mathbb{V}[\tilde{f}_i(p_i)]}{\mathbb{V}[\tilde{f}(\mathbf{P})]}, \quad S_{ij} = \frac{\mathbb{V}[\tilde{f}_{ij}(p_i, p_j)]}{\mathbb{V}[\tilde{f}(\mathbf{P})]}, \quad S_I = \frac{\mathbb{V}[\tilde{f}_I(p_I)]}{\mathbb{V}[\tilde{f}(\mathbf{P})]}. \tag{5a}$$

These quantities are known as first-order, second-order and higher-order Sobol sensitivity indices, respectively, and represent fractional sensitivities in the sense that they add up to unity,

$$1 = \sum_{i=1}^{N} S_i + \sum_{1 \leq i < j \leq N} S_{ij} + \cdots + S_{1,2,\dots,N}.$$

Finally, the "total effect" of a parameter $p_i$ is quantified by the so called Sobol total-effect index, defined as the sum of all sensitivity indices related to this parameter and its possible interactions,

$$S_{T_i} = S_i + \sum_{j \neq i} S_{ij} + \sum_{\substack{j \neq i, k \neq i \\ j < k}} S_{ijk} + \cdots + S_{1,2,\dots,N}. \tag{5b}$$

The Sobol sensitivity indices (5) are estimated via quasi-random sampling. To achieve this, the parameter space is sampled using a quasi-Monte Carlo method, which generates the so-called "Sobol sequences" – sequences of low-discrepancy quasi-random numbers designed to fill space uniformly [31]. The values of the function $\tilde{f}$ and the terms of its functional decomposition (3) are evaluated at these sequences. Finally, the "Jansen" estimator [32] is used to calculate the Sobol sensitivity indices from the sampled data. For this study, we employ an implementation of this procedure available from the SALib Sensitivity Analysis Library in Python [33]. Further technical details are omitted here, except to note that the consistency and bias of index estimation depend on the number $M$ of randomly sampled points in the parameter space. This dependence is investigated further below.

Sobol index estimates are random values drawn from a sampling distribution. To assess their uncertainty, bootstrapping is used [34]. The original sample is resampled with replacement 1000 times, generating bootstrap samples from which Sobol indices are recalculated. This process constructs empirical sampling distributions, providing 95% confidence intervals for the indices and assessing variability due to finite sample size $M$. Bootstrapping is preferred over repeated Monte Carlo estimations, which are computationally more expensive.

## Parameter ranking and dimensionality reduction

Sensitivity studies lead to a rank of sensitivity parameters in descending order of their influence on the model output. The ranking can be further exploited to reduce the dimensionality of the parameter space of the problem.

A number of different criteria for parameter ranking may be employed. In this study we define a "grand" total Sobol sensitivity index for each of the sensitivity parameters $p_i$, as

$$S_{Gi} = \sum_{k=1}^{K} w_k S_{Ti}^k. \tag{5c}$$

Here $S_{Ti}^k$ is the total Sobol index given by Eq (5b) which measures the overall influence of the parameter $p_i$ on a single AP biomarker $y^k$. In contrast the grand total index $S_{Gi}$ measures the overall influence of the parameter $p_i$ on all AP biomarkers $\mathbf{y}$. The weights $w_k$ can be used to tune the importance of the outputs from any a priori considerations; here we take $w_k = 1, k = 1, \dots, K$ as we consider all biomarkers to be of equal interest. The sensitivity parameters can then be ranked in descending influence by sorting the values of their grand total Sobol sensitivity indices. In the following, we denote parameter vectors where the components are ordered in decreasing grand total Sobol index by angular brackets

$$\langle \mathbf{p} \rangle = \langle p_1, p_2, \dots, p_N \rangle, \qquad S_{G1} \geq S_{G2} \geq \dots \geq S_{GN}, \tag{6}$$

and distinguish it from parameter vectors $\mathbf{p}$ where the order of components is immaterial.

If parameters are independent, the dimensionality of the parameter space can be reduced by keeping the values of a subset of $N-M$ parameters ($M \in [0, N]$) fixed to "hard-coded" constants. The natural choice is to fix the parameters with smallest grand total Sobol sensitivity indices as their influences on the model output are relatively less significant. Whether this is an acceptable approximation can be measured at any fixed point in the parameter space by the relative difference/error

$$e^{\langle M \rangle} = \frac{1}{K} \sum_{k=1}^{K} \left| \frac{(y_j^{\langle M \rangle, k} - y_j^k)}{y_j^k} \right|, \tag{7a}$$

between the output vectors of the reduced and the "full" models given by

$$\mathbf{y}^{\langle M \rangle} = \mathbf{f}(\langle p_1, p_2, \dots p_M, 1, \dots, 1 \rangle), \tag{7b}$$

$$\mathbf{y} = \mathbf{f}(\langle p_1, p_2, \dots p_M, p_{M+1}, \dots, p_N \rangle), \tag{7c}$$

respectively.

To obtain a global measure of the discrepancy over the entire parameter space we use quasi random Monte-Carlo sampling (similarly to assessing variability in section ), and compute the mean relative error from the generated samples of parameter vector values

$$E^{\langle M \rangle} = \frac{1}{LK} \sum_{j=1}^{L} \sum_{k=1}^{K} \left| \frac{(y_j^{\langle M \rangle, k} - y_j^k)}{y_j^k} \right|, \tag{8}$$

where $L = 2000$ is the number of samples. This has the advantage over a mean squared error or over a Euclidian norm that it can be interpreted directly as the relative average discrepancy between full and reduced models. When $E = 0$ agreement is perfect. Using the relative error has the further advantage of non-dimensionalising the errors of individual components which are otherwise measured in different units and may have significantly different magnitudes.

## Results and discussion: Sensitivity analysis of the Shannon model

In the following we present and discuss the results of applying the Sobol sensitivity analysis for a standard setup where the the stimulus amplitude was set to 9.5 A/F and trains of 1000 APs were generated with $t_{BCL}$ = 500 ms. Biomarkers were measured in the final AP from each train. Sample sequences of such measurements with size $M$ = 8192 were used to then evaluate the sensitivity indices. We investigated the effects of varying these standard setup choices, and found that the results and conclusions do not differ significantly. As an illustration, the percentage differences in biomarker values from their respective values measured at the 2000th beat are plotted in Fig 1 as a function of the length of the train, and show that trains of 1000 APs are sufficiently long to reach the steady state in the simulations even though rare random fluctuations smaller than 2% may occur occasionally due to accumulation of numerical error. The other standard setup choices are also justified further below.

### The parameter range of normal response

To fulfill their physiological functions, biological cells exhibit a variety of complex behaviors under different conditions. Similarly, mathematical models of the AP, such as the Shannon model (1), have qualitatively distinct solutions across their parameter space due to their non-linear nature [35]. The simplest response to periodic stimulation $I_{stim}(t)$ is a rapid return to a stable equilibrium (resting state), known as a 1:0 response, which mimics non-excitable cells. At other parameter values, a bifurcation to a periodic solution occurs, where each stimulus elicits a single AP, producing a 1:1 response that models normal physiological behaviour. Secondary bifurcations from the 1:1 response can result in a 2:1 response, where one stimulus generates an AP while the next does not, or a 2:2 response, where alternating stimuli

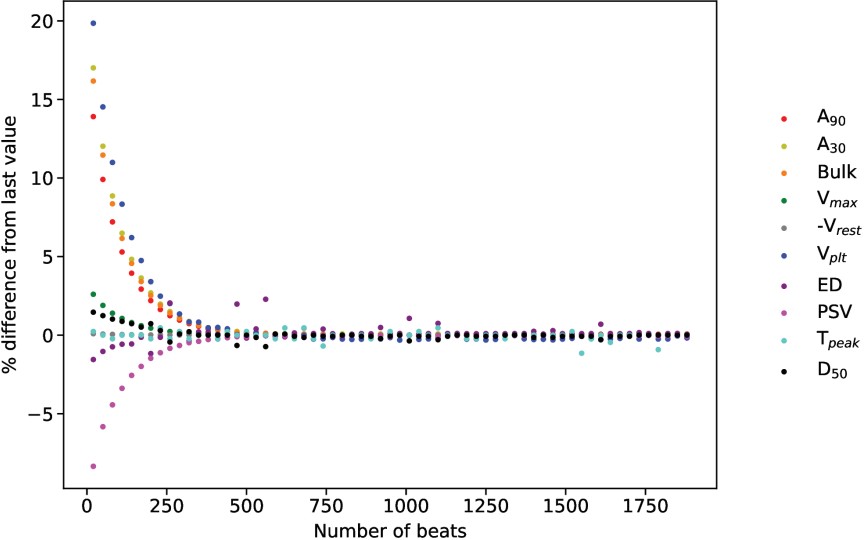

**Fig 1. Biomarkers as a function of the length of AP train.** The percentage differences in biomarker values from their respective values measured in the 2000th AP are plotted for increasing number of pre-pacing beats. The simulations are performed at 50% block of $I_{NaK}$ and 70% block of $I_{NaCa}$ and all other parameters at baseline values. Biomarkers are specified in the figure legend.

produce long and short APs in a stable 2-periodic pattern. This 2:2 response, known as alternans, is thought to reflect early signs of instability. Further tertiary instabilities can lead to chaotic regimes, representing abnormal electrophysiological behavior, which may culminate in fibrillation. The review article of [35] discusses nonlinear and stochastic dynamics in the heart.

Naturally, sensitivity to parameters is expected to vary significantly across different solution regimes. We restrict our sensitivity analysis of the Shannon model to the parameter region corresponding to the normal 1:1-response, as this represents the most common physiological behavior. To perform this analysis, it is first necessary to identify the region in the parameter space where the primary normal response occurs. This region, occasionally called a "Busse balloon" in the context of pattern-forming dynamical systems [36], is not the primary focus of this study. Instead, our goal is to identify a sufficiently large region of normal response suitable for sensitivity analysis. To achieve this, we begin with the baseline Shannon model, where $\{p_i = 1\}_{i=1}^N$, as the baseline model is calibrated to produce a normal response. Each sensitivity parameter $p_i$ is then varied independently within the range $10^{-4}$ to 10, until the first transition to a secondary instability or a non-excitable state is observed.

Fig 2 illustrates the procedure for the parameter $p_{\text{NaK}}$. One biomarker, the AP duration at 90% repolarization ($A_{90}$), is used to assess the model's response, as other biomarkers exhibit similar behavior. At $p_{\text{NaK}} = 1$, a normal 1:1-response is recorded, as shown in Fig 2(d). As a side note: Fig 2(d) which also depicts a typical AP profile in this model. The normal response persists for $p_{\text{NaK}}$ in the range $[10^{-4}, 1.07]$. Beyond $p_{\text{NaK}} = 1.07$, an intermittent response emerges, where approximately 20 normal APs are followed by an equally long equilibrium phase, as illustrated in Fig 2(e). This behavior produces a two-valued curve in the $(p_{\text{NaK}}, A_{90})$ plane, with $A_{90} \approx 0$ during equilibrium and finite values during the normal response sequence. For $p_{\text{NaK}} > 2.1$, only the equilibrium response is observed, as shown in Fig 2(f). Table 4 summarizes these findings and defines the parameter subspace for the normal response region analyzed in this study.

The threshold of excitation at baseline parameter values is approximately 9.4 A/F, so our standard setup uses a slightly supercritical stimulus amplitude. Increasing the amplitude of the stimulus current enlarges the region of normal response as shown in Fig 2(a), 2(b), 2(c) where results for amplitudes up to 2 times the threshold are also included. However, increasing the stimulus amplitude does not significantly alter the sensitivity ranking of parameters as will be shown further below.

## Consistency and bias of Sobol's index estimators

The Sobol sensitivity indices, as statistical estimators derived from random samples (Section ), must exhibit key properties of good estimators, such as consistency (convergence to true values with larger sample sizes) and unbiasedness (no systematic deviation from true values). Here, we evaluate these properties to ensure the robustness of our Sobol sensitivity indices estimation.

Fig 3 demonstrates the consistency of first-order and total-order indices of the parameter $p_{\text{Clb}}$ in the large-sample limit across the six AP biomarkers. Specifically, smaller sample sizes $M$ show greater variability, particularly in the total-order indices which are sensitive to interaction effects. As the sample size $M$ is increased, both first-order and total-order indices converge to specific values. The sensitivity indices of all other parameters behave similarly. Based on this consistency test we have fixed the sample size to $M = 8192$ in our subsequent analysis.

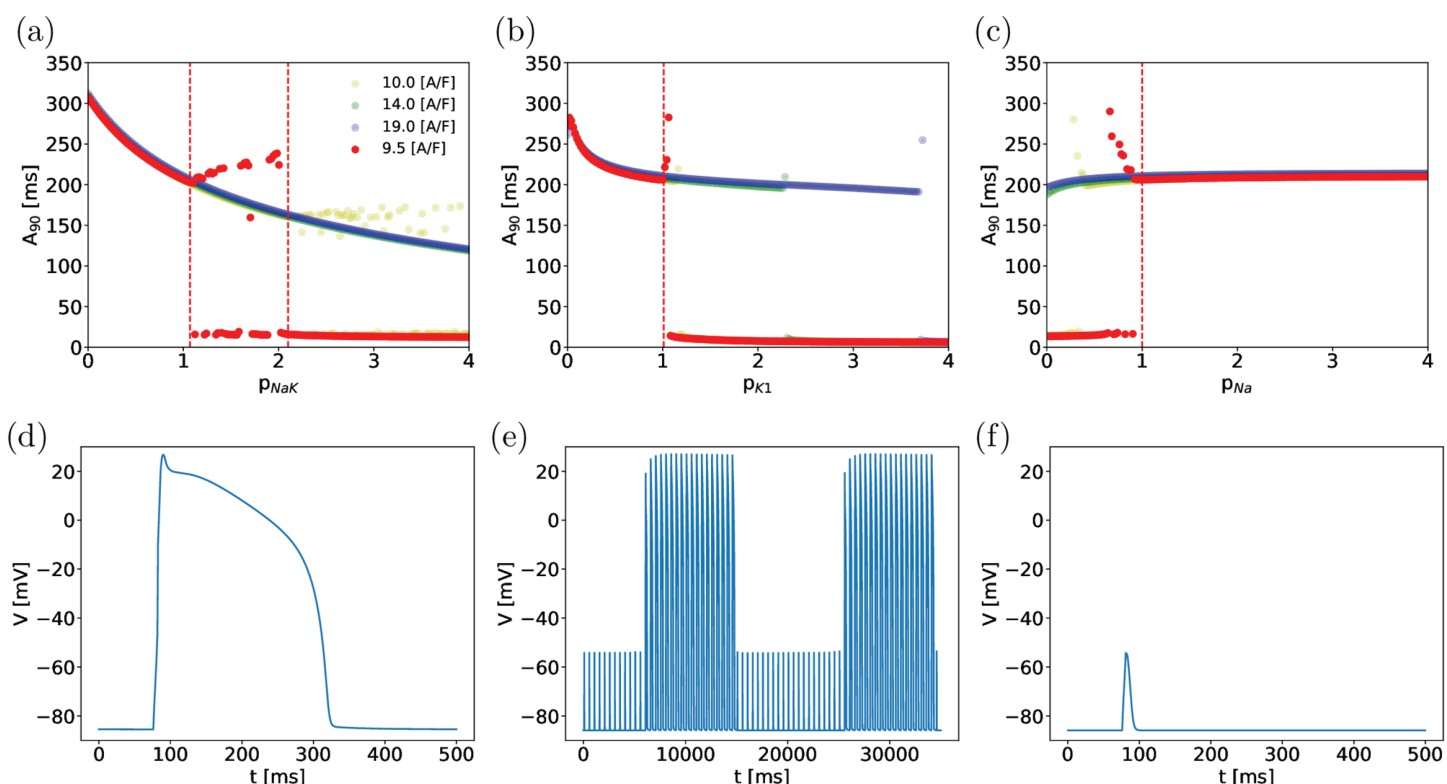

**Fig 2. (Top row) Ranges of parameter values for normal response illustrated on a selection of parameters and for various stimulus amplitudes.** Values of $A_{90}$ as functions of $p_{NaK}$, $p_{K1}$ and $p_{Na}$ in (a), (b) and (c), respectively, and for four stimulus amplitudes as listed in the legend. The red dashed lines outline regime boundaries (for amplitude 9.5 A/F) as discussed in the text. (Bottom row) Types of response to stimuli observed for various values of the maximal density of the sodium-potassium pump current $p_{NaK}$ for amplitude 9.5 A/F. (d): Normal response where each stimulus elicits a single 1:1 AP is observed for $p_{NaK} < 1.07$; (e): An "intermittent" excitation pattern where $Q$ successive stimuli elicit $Q$ normal APs, while the next $Q$ stimuli do not, observed for $p_{NaK} \in (1.07, 2.1)$ (f): Stimuli do not elicit an AP response for $p_{NaK} > 2.1$.

**Table 4. Parameter subspace of normal response. Sensitivity analysis is performed within this range of parameter values.**

| Parameter | Range | Parameter | Range | Parameter | Range |
|---|---|---|---|---|---|
| $p_{NaK}$ | [0.0001, 1.07] | $p_{Kr}$ | [0.0001, 3.2] | $p_{tos}$ | [0.0001, 2.5] |
| $p_{K1}$ | [0.03, 1.01] | $p_{NaCa}$ | [0.6, 3] | $p_{Clb}$ | [0.0001, 10] |
| $p_{CaL}$ | [0.7, 1.6] | $p_{Ks}$ | [0.0001, 10] | $p_{Cab}$ | [0.34, 10] |
| $p_{Cap}$ | [0.0001, 4.5] | $p_{ClCa}$ | [0.0001, 10] | $p_{Kp}$ | [0.0001, 10] |
| $p_{Na}$ | [1, 10] | $p_{Nab}$ | [0.0001, 10] | $p_{tof}$ | [0.0001, 4] |

To assess the bias of the Sobol sensitivity indices, we compute bootstrap distributions of the first-order and total-order indices, revealing their variability and central tendency. Comparing the mean of the bootstrap estimates to the original Sobol indices allows detection of systematic deviations and bias if any. Fig 4 shows the bootstrap distributions for $p_{Clb}$ with the AP feature $A_{90}$. Both first-order and total-order indices exhibit normal distributions, confirming unbiased estimates at $M = 8192$. Similar unbiased behavior is observed for all other parameters.

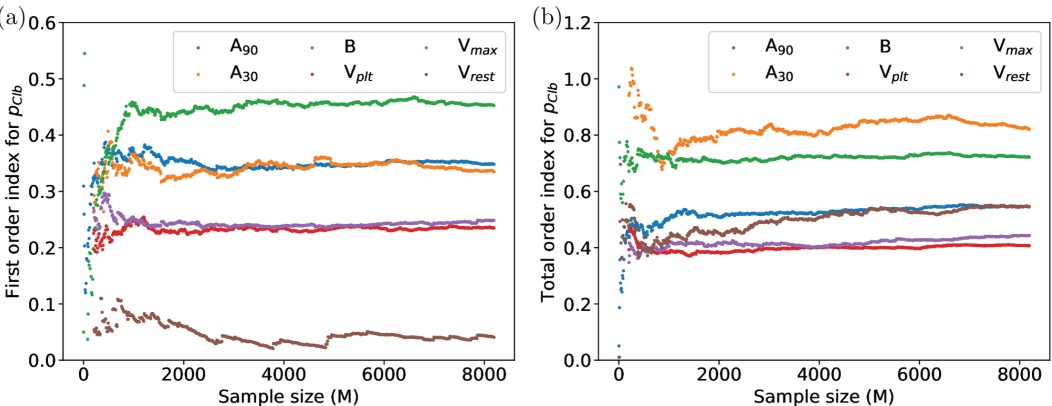

**Fig 3. Consistency of Sobol sensitivity index estimation.** Values of the first-order (a) and total-order (b) sensitivity indices of the background chloride current maximal density $p_{Clb}$ for the six AP biomarkers of interest (as stated in the legend) when calculated with different numbers of samples.

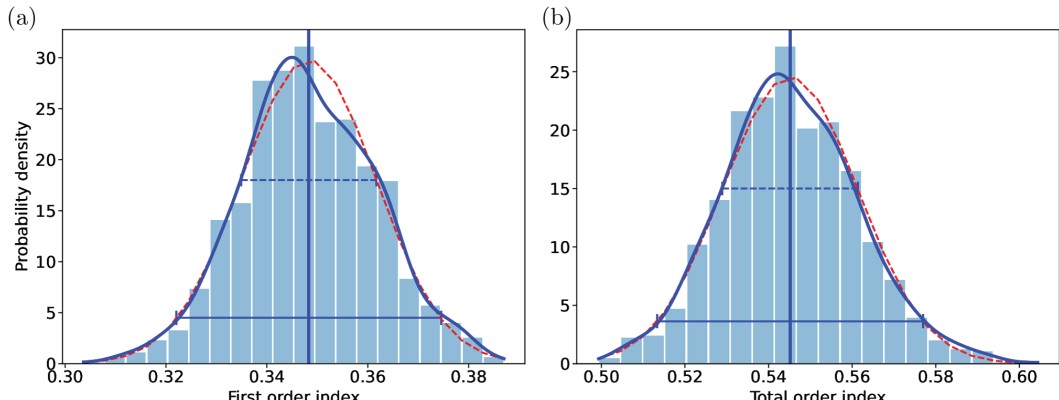

**Fig 4. Bias of Sobol sensitivity index estimation.** Histograms of the bootstrapping distributions of the first-order (a) and total-order (b) sensitivity indices of the background chloride current maximal density $p_{Clb}$ for $A_{90}$ constricted with a sample size of $M = 8192$ are shown in blue. The mean value and the standard deviation of the bootstrapping histograms are shown by the vertical blue line and the horizontal blue bar, respectively. The latter are used to plot a normal distribution (red dashed line) for comparison. The kernel density estimates (KDE) bootstrapping distributions are shown as blue lines. Horizontal blue bars show 1 (broken line) and 2 (solid) standard deviations from mean of the bootstraping distributions.

Bootstrap distributions, such as those in Fig 4, provide a basis for constructing confidence intervals for the Sobol sensitivity indices. Using the empirical "68–95–99.7" rule, approximately 95% of estimates lie within two standard deviations of the mean. Confidence interval lengths computed in this way are presented in Tables 5 and 6 for all Sobol sensitivity index estimates.

## The most and the least important ionic currents in Shannon's model

We are now ready to rank the relative strengths of ionic currents in the Shannon model based on their influence on the model output biomarkers. The relative strengths of ionic currents $I_i$, compared to their baseline values from [4], are represented by the factors $p_i$, while the output biomarkers are listed in Table 3. The importance of each current is quantified using the grand

**Table 5. The top three first-order and total-order sensitivity indices with corresponding confidence intervals for the six AP biomarkers considered.** $S_i^k$ is the first Sobol index given by Eq (5a) which measures the individual influence of the parameter $p_i$ on a single AP biomarker $y^k$ and $S_{Ti}^k$ is the total Sobol index given by Eq (5b) which measures the overall influence of the parameter $p_i$ on a single AP biomarker $y^k$.

| $y^k$ | parameter | $S_i^k$ | 95% CI | parameter | $S_{Ti}^k$ | 95% CI |
|---|---|---|---|---|---|---|
| $A_{30}$ | $p_{Clb}$ | 0.33 | 0.032 | $p_{Clb}$ | 0.82 | 0.065 |
| | $p_{NaCa}$ | 0.063 | 0.019 | $p_{Kr}$ | 0.49 | 0.056 |
| | $p_{ClCa}$ | 0.036 | 0.012 | $p_{NaCa}$ | 0.47 | 0.05 |
| $A_{90}$ | $p_{Clb}$ | 0.35 | 0.023 | $p_{Clb}$ | 0.54 | 0.036 |
| | $p_{Kr}$ | 0.14 | 0.019 | $p_{Kr}$ | 0.34 | 0.023 |
| | $p_{NaCa}$ | 0.14 | 0.016 | $p_{NaCa}$ | 0.24 | 0.018 |
| $B$ | $p_{Clb}$ | 0.45 | 0.027 | $p_{Clb}$ | 0.72 | 0.03 |
| | $p_{NaCa}$ | 0.076 | 0.015 | $p_{Kr}$ | 0.26 | 0.025 |
| | $p_{CaL}$ | 0.031 | 0.011 | $p_{NaCa}$ | 0.24 | 0.023 |
| $V_{plt}$ | $p_{Clb}$ | 0.23 | 0.021 | $p_{Clb}$ | 0.41 | 0.022 |
| | $p_{K1}$ | 0.17 | 0.020 | $p_{K1}$ | 0.35 | 0.025 |
| | $p_{CaL}$ | 0.15 | 0.017 | $p_{Kr}$ | 0.26 | 0.025 |
| $V_{max}$ | $p_{CaL}$ | 0.32 | 0.024 | $p_{Clb}$ | 0.44 | 0.030 |
| | $p_{Clb}$ | 0.25 | 0.024 | $p_{CaL}$ | 0.40 | 0.022 |
| | $p_{tos}$ | 0.050 | 0.013 | $p_{K1}$ | 0.14 | 0.0098 |
| $V_{rest}$ | $p_{K1}$ | 0.23 | 0.026 | $p_{Clb}$ | 0.55 | 0.054 |
| | $p_{Kr}$ | 0.068 | 0.028 | $p_{Kr}$ | 0.54 | 0.052 |
| | $p_{Clb}$ | 0.041 | 0.030 | $p_{K1}$ | 0.47 | 0.042 |

**Table 6. Selected second-order sensitivity indices of 18 parameter pairs for 4 AP biomarkers.** $S_{ij}^k$ is the second Sobol index given by Eq (5a) which measures the pairwise interaction influence of the parameters $p_i$ and $p_j$ on a single AP biomarker $y^k$. Similar with the work by [37], only the parameter pairs with positive lower limit of the CI are included, showing their significant influences on AP biomarkers.

| $y^k$ | Parameter pair | $S_{ij}^k$ | 95% CI | $y^k$ | Parameter pair | $S_{ij}^k$ | 95% CI |
|---|---|---|---|---|---|---|---|
| $A_{90}$ | $(p_{Kr}, p_{Clb})$ | 0.089 | 0.039 | $B$ | $(p_{Kr}, p_{Clb})$ | 0.032 | 0.031 |
| $B$ | $(p_{Kr}, p_{Ks})$ | 0.029 | 0.021 | $B$ | $(p_{Ks}, p_{Cap})$ | 0.024 | 0.016 |
| $B$ | $(p_{Ks}, p_{ClCa})$ | 0.019 | 0.018 | $B$ | $(p_{Ks}, p_{ClCa})$ | 0.019 | 0.018 |
| $B$ | $(p_{Ks}, p_{Kp})$ | 0.023 | 0.016 | $B$ | $(p_{Ks}, p_{Na})$ | 0.023 | 0.017 |
| $B$ | $(p_{Ks}, p_{Nab})$ | 0.022 | 0.017 | $B$ | $(p_{Ks}, p_{tof})$ | 0.024 | 0.017 |
| $B$ | $(p_{Kp}, p_{Na})$ | 0.0087 | 0.0066 | $B$ | $(p_{Kp}, p_{Nab})$ | 0.0084 | 0.0063 |
| $B$ | $(p_{Kp}, p_{tof})$ | 0.0098 | 0.0065 | $B$ | $(p_{Na}, p_{tof})$ | 0.0055 | 0.0052 |
| $V_{rest}$ | $(p_{Kr}, p_{Clb})$ | 0.11 | 0.059 | $V_{rest}$ | $(p_{NaCa}, p_{ClCa})$ | 0.026 | 0.024 |
| $V_{rest}$ | $(p_{NaCa}, p_{Kp})$ | 0.024 | 0.022 | $V_{max}$ | $(p_{K1}, p_{Clb})$ | 0.028 | 0.019 |

total Sobol sensitivity index, defined in Eq (5c), which measures the combined effect of a single parameter $p_i$ on all biomarkers. The grand total Sobol indices for all 15 currents are shown in Fig 5, ranked in decreasing order of their influence on model outputs. We find that, in the notation of Eq (6) and for stimulus amplitude 9.5 A/F, the vector of ordered parameters is

$$\langle p_{Clb}, p_{Kr}, p_{K1}, p_{CaL}, p_{NaCa}, p_{Ks}, p_{NaK}, p_{tos}, p_{ClCa}, p_{Cab}, p_{tof}, p_{Cap}, p_{Nab}, p_{Na}, p_{Kp} \rangle. \tag{9}$$

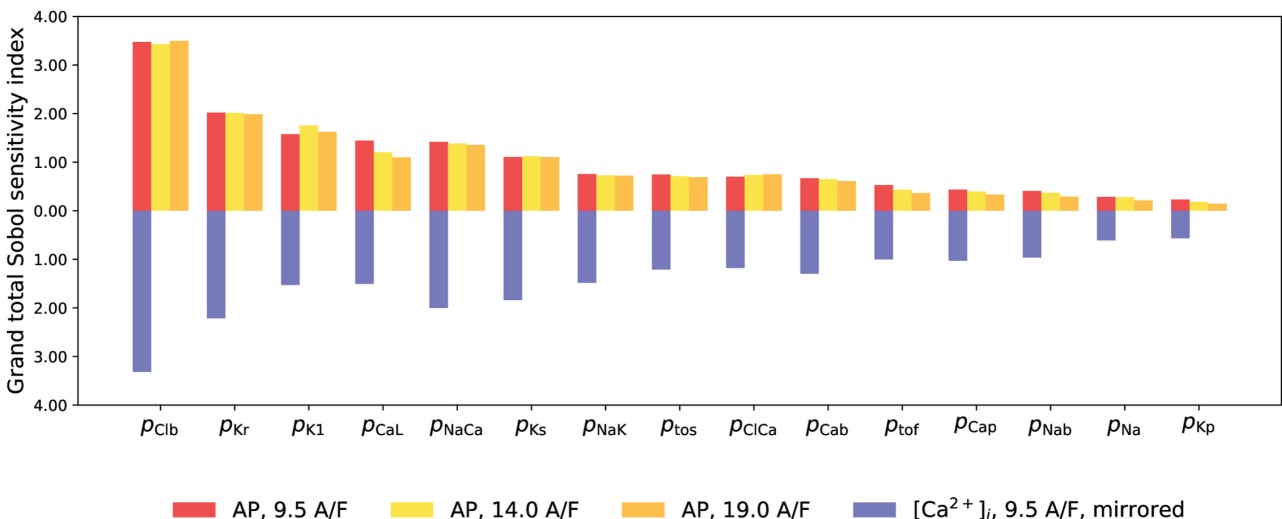

**Fig 5. Global sensitivity ranking of the parameters considered in the analysis.** Barplots of the grand total Sobol sensitivity index with respect to AP biomarkers for three different stimulus amplitudes, as well as with respect to $[Ca^{2+}]_i$ biomarkers (for amplitude 9.5 A/F only) are included as indicated in the legend.

Increasing the stimulus amplitude up to 2 times the threshold value does not significantly affect the ranking as also shown in Fig 5. The relative strength of the background chloride current $p_{Clb}$, emerges as the most influential parameter in the Shannon model, followed by that of the potassium current $p_{Kr}$, the inward rectifier potassium current $p_{K1}$, the L-type calcium current $p_{CaL}$, the sodium-calcium exchanger current $p_{NaCa}$, and the slow delayed rectifier potassium current $p_{Ks}$. On the other hand, the relative strength of the background potassium current $p_{Kp}$ is found to be the least influential parameter in the model. This rank list constitutes the central result of our analysis.

The results summarised in Fig 5 are in general agreement with published experimental findings in terms of identifying the major conductances that influence repolarization of the cardiac AP. A surprising observation emerged from the ranking of global sensitivity is that the highest sensitivity is attributed to the value of the conductance of the background chloride current ($I_{Clb}$). While this was not expected, a large sensitivity to $I_{Clb}$ was also independently reported in the work of [?] for the Grandi human atrial AP model [39] which in turn is based on the Shannon model. One possible explanation may be that because the intracellular concentration of Cl is constant in the definition of the Shannon model the alteration of this current does not influence this concentration in simulations, but in reality this concentration could be significantly altered when varying $I_{Clb}$. Our sensitivity analysis reflects the behaviour of the model as currently formulated. The importance of $I_{Clb}$ seems to be generally underappreciated as noted by [40] despite its importance in determining AP duration and the resting membrane potential, both key parameters determining the electrical stability of the heart [41]. In contrast, the next most influential current, the rapidly inactivating delayed rectifier potassium current ($I_{Kr}$), carried by the hERG channel, is considerably more intensively studied due to the linkage to the pro-arrhythmic condition associated with long QT. The activation of $I_{Kr}$ current late in the repolarization phase ensures the rapid restoration of the resting membrane potential [42,43] and reduced $I_{Kr}$ prolongs the AP and therefore the QT duration. The slowly inactivating delayed rectifier potassium current ($I_{Ks}$) contributes to the cardiac repolarization, particularly during increased heart rates, by preventing excessive prolongation

of the AP [44]. The inwardly rectifying potassium current ($I_{K1}$) is important in determining the resting membrane potential, the initial depolarization and the final repolarization phases of the AP [45,46]. Early repolarization (phase 1) is shaped by the transient outward potassium current ($I_{tos}$), which contributes to the characteristic notch in the AP [47]. The L-type calcium current ($I_{CaL}$), contributes to the initial depolarization (phase 0) initiated by the activation of inward sodium current by contributing to the maximum positive potential [48]. Sustained inward calcium current during the plateau of the AP (phase 2) helps to maintain depolarized potentials and therefore the overall AP duration [48]. The magnitude and direction of the sodium-calcium exchanger current ($I_{NaCa}$) is dependent on the intracellular sodium and calcium concentrations and the transmembrane voltage [49]. At peak systolic calcium, the high sub-sarcolemma calcium concentrations ensures that the $I_{NaCa}$ is an inward current, causing net efflux of calcium from the cell and contributing to the plateau phase of the AP. On repolarization due to activation of delayed rectifier currents, $I_{NaCa}$ remains an inward current and is one of the main calcium efflux mechanisms [50]. The sodium-potassium pump current ($I_{NaK}$) is a consequence of the electrogenic stoichiometry of the pump that actively extrudes sodium while importing potassium, maintaining intracellular ion homeostasis. The $I_{NaK}$ magnitude is normally small compared to the other currents listed above and therefore the direct influence on the AP is small. But the activity of the sodium potassium pump maintains normal intracellular sodium levels and thereby indirectly influences intracellular calcium via the sodium calcium exchanger [49]. The indirect effects on calcium sensitive currents and the effects of altered intracellular potassium and sodium concentrations on ionic currents explain the significant indirect influence of the sodium potassium pump on the AP waveform [51].

To produce a secondary ranking of the ionic currents based on their influence on individual biomarkers, we analyze the corresponding total, first-order, and second-order indices. The total and first-order indices are shown in the left columns of Fig 6. The total-order index, defined in Eq (5b), quantifies a parameter's contribution to the variability of a single biomarker, accounting for both direct impacts and interactions with other parameters. The background chloride current relative strength parameter, $p_{Clb}$, is consistently the most influential for all action potential biomarkers. Following $p_{Clb}$, the rapid delayed rectifier potassium current parameter ($p_{Kr}$), the inward rectifier potassium current parameter ($p_{K1}$), the sodium-calcium exchange current parameter ($p_{NaCa}$), and the L-type calcium current parameter ($p_{CaL}$) exhibit significant total-order sensitivity.

To determine whether specific ionic currents influence a biomarker individually or through interactions with other currents, we analyze their first-order Sobol indices. Defined in Eq (5), the first-order index quantifies the direct effect of a single parameter on a single biomarker, representing the proportion of variance caused solely by that parameter. The first-order indices for the relative strengths of ionic currents are shown in the left column of Fig 6. The background chloride current parameter ($p_{Clb}$) is consistently dominant, particularly for biomarkers such as action potential duration at 30% and 90% repolarization ($A_{30}$ and $A_{90}$), plateau voltage, and AP bulk. For example, $p_{Clb}$ contributes five times more to the variance of $A_{30}$ than the second- and third-ranked parameters. The resting membrane potential ($V_{rest}$) is most sensitive to the potassium current parameters ($p_{K1}$ and $p_{Kr}$), while the maximum AP upstroke ($V_{max}$) is strongly influenced by the L-type calcium current parameter ($p_{CaL}$) as it is the major depolarizing current during late upstroke. This effect of calcium current is through the effect on ($V_{max}$), not influencing the maximum upstroke velocity as ($p_{Na}$) is almost the sole determinant parameter [52]. The sodium-calcium exchange current parameter ($p_{NaCa}$) consistently ranks among the top three for several biomarkers, highlighting its importance in action potential dynamics. Conversely, the background potassium current parameter ($p_{Kp}$)

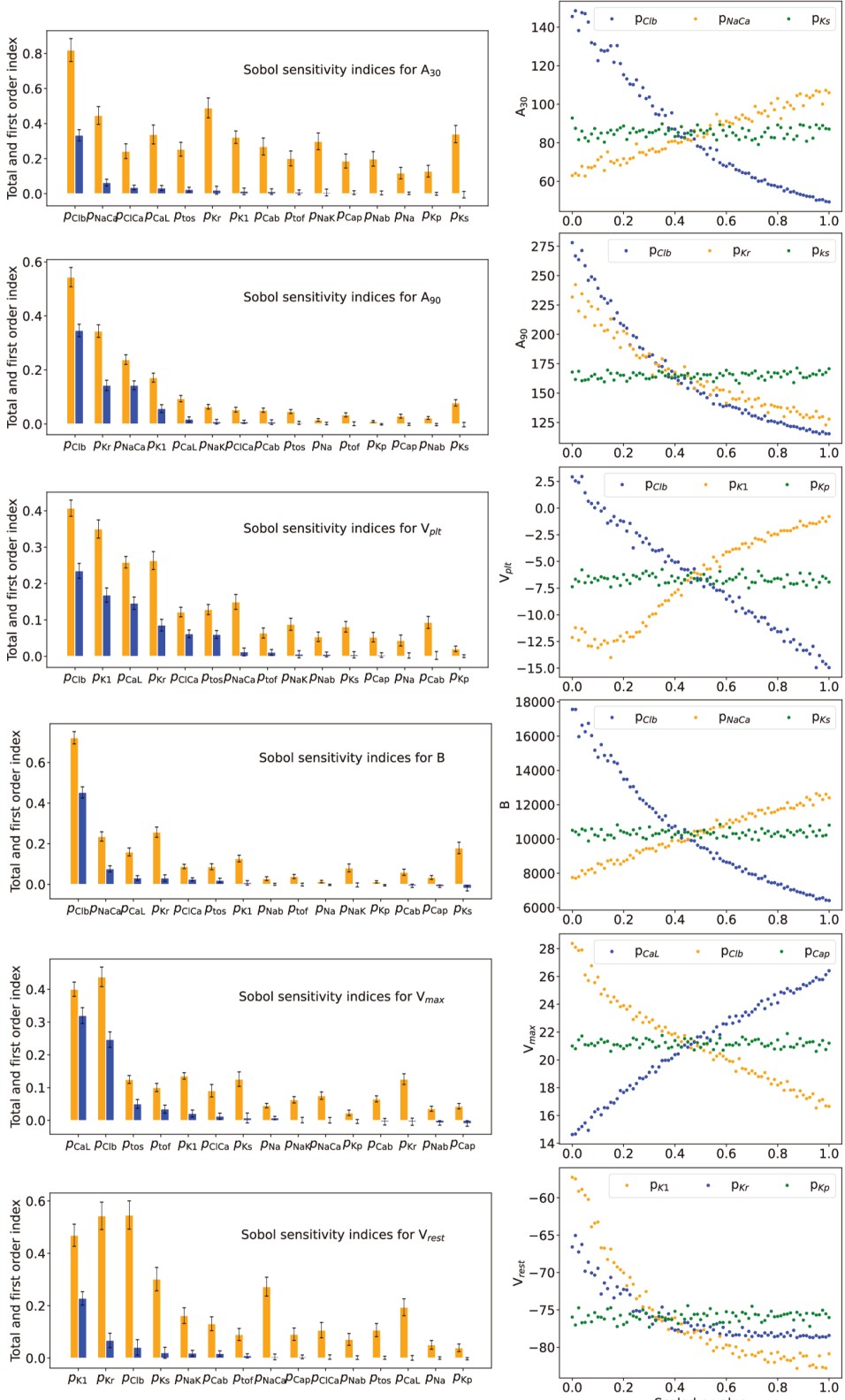

**Fig 6. (Left column) Total and first-order Sobol sensitivity indices for the six studied biomarkers sorted in descending order with respect to the first-order index.** (Right column) Conditional expectations for the parameters with the largest, the second-largest and the smallest values of the first-order Sobol sensitivity index. To aid visualisation parameter values $p$ are scaled by $p_{scaled} = (p - p_{min})/(p_{max} - p_{min})$ to [0,1] on the abscissa.

and the slow delayed rectifier potassium current parameter ($p_{Ks}$) exhibit the lowest first-order indices across all biomarkers.

To assess the joint influence of two currents on biomarkers, we examine their second-order Sobol indices. Table 6 lists all second-order indices with a positive lower confidence interval limit. Notably, $S_{Kr,Clb}$ emerges as the highest second-order index for $A_{90}$, $B$, and $V_{rest}$, while $S_{K1,Clb}$ dominates for $V_{max}$. The strong interaction between $Cl^-$ and $K^+$ conductances aligns with findings in [41], which suggest that the role of $I_{Clb}$ in AP waveform variability depends on its interplay with other ion conductances, especially $K^+$. Reducing extracellular potassium concentration to suppress $I_{Kr}$ and $I_{K1}$ increases resting potential depolarization, amplifying the depolarizing action of $I_{Clb}$ [41]. Additionally, the slow delayed rectifier potassium current parameter ($p_{Ks}$) shows significant second-order interaction effects on the AP bulk ($B$), particularly with parameter $p_{Kr}$. This supports findings by [53] that blocking $I_{Ks}$ markedly prolongs AP duration when "repolarization reserve" is reduced by $I_{Kr}$ block.

Having identified the parameters that most strongly influence the AP biomarkers, we next explore the explicit relationships between them. The second column of Fig 6 displays the conditional expectations of the biomarkers with respect to the parameters with the highest and lowest first-order Sobol indices. These plots show the expected value of a biomarker as a function of a parameter while all other parameters vary randomly. These conditional expectations provide insights into the general dependence of biomarkers on specific parameters and indicate how the biomarker is likely to vary as one parameter changes while the values of all others remain uncertain.

The work of [16] highlights the significant influence of repolarization currents, particularly $p_{CaL}$, $p_{Kr}$, $p_{tos}$, $p_{NaK}$, and $p_{NaCa}$, on AP waveform. Consistent with this, we find these parameters ranked within the top eight in Fig 5. Unlike [16], our analysis reveals $p_{Ks}$ as influential, likely due to differences in methodology. Their local sensitivity method varied selected parameters by only $\pm 15\%$ and $\pm 30\%$ from the Shannon model's baseline values [4], limiting the analysis to a narrow parameter neighborhood and ignoring interaction effects. In contrast, our global sensitivity method explores a wide parameter space, allowing all parameters to vary simultaneously. This broader approach captures interaction effects, as illustrated in Fig 6, where $G_{Ks}$ exhibits low first-order indices for six AP biomarkers but high total-order indices, indicating strong higher-order interaction effects on the AP waveform.

The work of [5] systematically explored the effects of simultaneously varying the magnitude of six transmembrane current conductances in the [4] model. Using clutter-based dimension reordering, they identified $p_{CaL}$ as having the greatest influence on AP variability at both 400 ms and 1000 ms, along with $p_{K1}$ and $p_{to}$ at 400 ms and 1000 ms, respectively. These findings align with our results, where these three parameters also ranked among the top eight in Fig 5, despite our analysis being conducted at a different basic cycle length.

In contrast to the findings of [10], $p_{Na}$ is not the most influential parameter in this study. This discrepancy may stem from our focus on successfully excited cells, which for stimulus amplitude 9.5 A/F limits the range of $p_{Na}$ to values $>1$. If values of $p_{Na} < 1$ were to be considered, the effects would have included the differences between normal and failing AP responses which are, of course, huge. The effects of $p_{Na}$ on normal APs alone are far less significant.

## Extension sensitivity analysis for intracellular calcium biomarkers

A number of studies using populations of models use biomarkers of intracellular calcium concentration to calibrate them [54,55]. To facilitate this, we have extended the Sobol sensitivity analysis to include the following four intracellular calcium biomarkers: (a) ED: $[Ca^{2+}]_i$

at the end of diastole; (b) PSV: peak systolic value of $[Ca^{2+}]_i$; (c) $T_{peak}$: time from stimulus to peak $[Ca^{2+}]_i$; (d) $D_{50}$: period of time when $[Ca^{2+}]_i$ remains elevated above a threshold of 50% recovery from the peak value to the resting value, informally duration at 50% amplitude. The grand total Sobol indices based on these four biomarkers are shown in blue in Fig 5 for the standard setup of our study. The top seven most influential parameters remain the same as those found by analysis of AP-biomarkers shown in red in Fig 5, with the relative strength of the background chloride current $p_{Clb}$ still emerging as the most influential parameter, followed by $p_{Kr}, p_{NaCa}, p_{Ks}$. On the other hand, the relative strength of the background potassium current $p_{Kp}$ remains to be the least influential parameter in the model. The higher rank of $p_{NaCa}$ reflects its large importance for intracellular calcium dynamics and this is because the rise in intracellular calcium concentration activates the Na+/Ca2+ exchanger cell membrane pump [56].

## A hierarchy of reduced Shannon models

Using the ranked Shannon model parameters based on their grand Sobol sensitivity indices (see Fig 5 and Eq (9)), we now address the second main goal of our sensitivity analysis: constructing a hierarchy of reduced Shannon models. This hierarchy is formed by successively fixing the least significant parameters to their baseline values, as follows:

$$
\begin{aligned}
\mathbf{y} &= \mathbf{f}\big(\langle p_{Clb}, p_{Kr}, p_{K1}, p_{CaL}, p_{NaCa}, p_{Ks}, p_{NaK}, p_{tos}, p_{ClCa}, p_{Cab}, p_{tof}, p_{Cap}, p_{Nab}, p_{Na}, p_{Kp}\rangle\big), \\
\mathbf{y}^{\langle 14\rangle} &= \mathbf{f}\big(\langle p_{Clb}, p_{Kr}, p_{K1}, p_{CaL}, p_{NaCa}, p_{Ks}, p_{NaK}, p_{tos}, p_{ClCa}, p_{Cab}, p_{tof}, p_{Cap}, p_{Nab}, p_{Na}, 1\rangle\big), \\
\mathbf{y}^{\langle 13\rangle} &= \mathbf{f}\big(\langle p_{Clb}, p_{Kr}, p_{K1}, p_{CaL}, p_{NaCa}, p_{Ks}, p_{NaK}, p_{tos}, p_{ClCa}, p_{Cab}, p_{tof}, p_{Cap}, p_{Nab}, 1, 1\rangle\big), \\
&\quad \cdots \\
\mathbf{y}^{\langle 1\rangle} &= \mathbf{f}\big(\langle p_{Clb}, 1, 1, 1, 1, 1, 1, 1, 1, 1, 1, 1, 1, 1\rangle\big).
\end{aligned}
\tag{10}
$$

The original Shannon model [4] corresponds to

$$
\mathbf{y}^{\langle 0\rangle} = \mathbf{f}\big(\langle 1, 1, 1, 1, 1, 1, 1, 1, 1, 1, 1, 1, 1, 1, 1\rangle\big), \tag{11}
$$

representing the lowest hierarchical level where all parameters are fixed.

Fig 7 quantifies the global discrepancy between the full model ($\mathbf{y}$) and the reduced models ($\mathbf{y}^{\langle M\rangle}$) at hierarchical levels $M$ over the parameter ranges of normal response. Two measures of discrepancy are presented: the mean relative error, defined by Eq (8) and plotted on the left-hand ordinate axis, and the coefficient of determination ($R^2$), shown on the right-hand ordinate axis, both as functions of $M$. For models $\mathbf{y}^{\langle 0\rangle}$ to $\mathbf{y}^{\langle 5\rangle}$, where fewer than the six most sensitive parameters are varied, the discrepancy is non-monotonic and remains too large. Consequently, these models do not approximate the full model well and are excluded from the plots in Fig 7, which only includes models from $\mathbf{y}^{\langle 6\rangle}$ to $\mathbf{y}$. At higher hierarchical levels ($M \geq 6$), the discrepancy decreases monotonically with increasing $M$. The mean relative error approaches zero, while $R^2$ approaches unity, indicating improved accuracy. Fig 8 provides a local comparison of the reduced and full models to offer an intuitive understanding of their accuracy.

While a hierarchy of models can be constructed by ranking parameters based on their grand total sensitivity index, more "economical" specialized reduced models can be obtained to capture individual biomarkers. For each biomarker of interest, this is achieved by ranking parameters based on their total sensitivity indices. To illustrate, Fig 9 presents scatter plots of $A_{90}$ values obtained from the reduced model

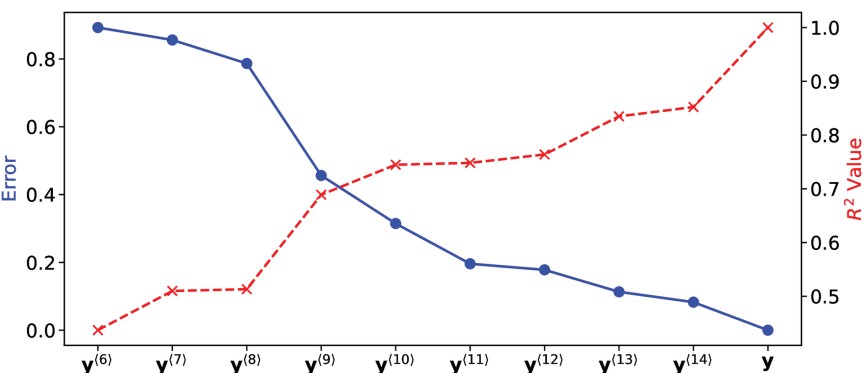

**Fig 7. Deviation of reduced models $\mathbf{y}^{(M)}$ from the full model $\mathbf{y}$ as measured by the mean relative error (blue dots; left ordinate axis) and the coefficient of determination $R^2$ (red crosses, right ordinate axis).**

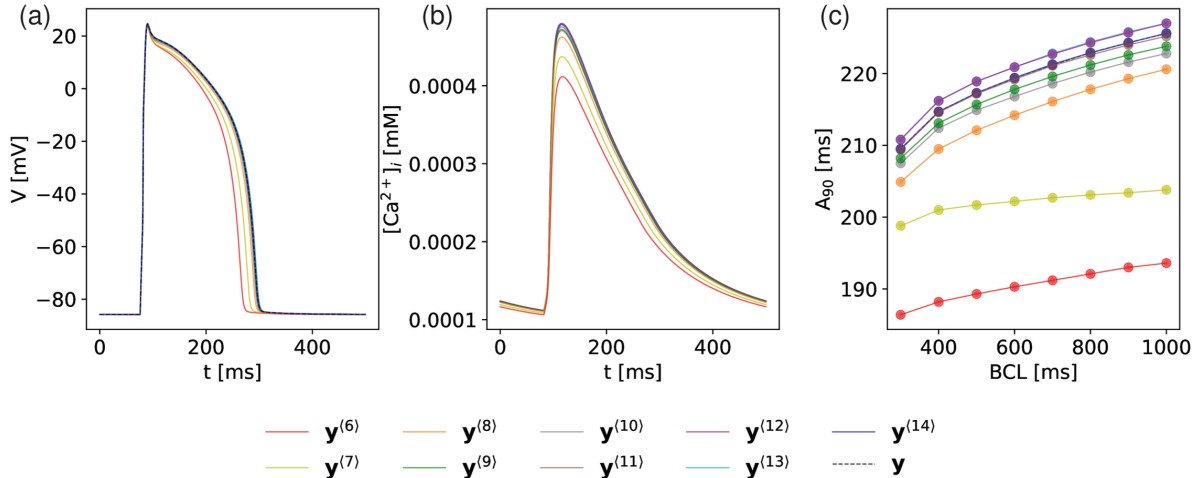

**Fig 8. A local illustration of the accuracy of reduced models. Panel (a) and (b) show AP forms, and Calcuim transients $[\mathrm{Ca}]_i^{2+}$, while panel (c) shows $A_{90}$ restitution curves ($A_{90}$ for periodically excited waveforms as a function of basic cycle length) for reduced models $\mathbf{y}^{(6)}$ to $\mathbf{y}^{(14)}$ in comparison with the full model $\mathbf{y}$.** The models are evaluated locally so that for the full model all parameters are set equal to 0.8, while for the models at hierarchical level $M$ the $M$ "variable" parameters are also set to 0.8 while the rest are kept at baseline value corresponding to 1.

$$\mathbf{y}^{(6)} = \mathbf{f}\Big(\big\langle p_{\mathrm{Clb}}, p_{\mathrm{Kr}}, p_{\mathrm{NaCa}}, p_{\mathrm{K1}}, p_{\mathrm{CaL}}, p_{\mathrm{Ks}}, 1, 1, 1, 1, 1, 1, 1, 1, 1\big\rangle\Big),$$

and its "complement" model

$$\overline{\mathbf{y}^{(6)}} = \mathbf{f}\Big(\big\langle 1, 1, 1, 1, 1, 1, p_{\mathrm{NaK}}, p_{\mathrm{tos}}, p_{\mathrm{ClCa}}, p_{\mathrm{Cab}}, p_{\mathrm{tof}}, p_{\mathrm{Cap}}, p_{\mathrm{Nab}}, p_{\mathrm{Na}}, p_{\mathrm{Kp}}\big\rangle\Big),$$

against the corresponding values from the full model in panels (a) and (b), respectively. The coefficient of determination ($R^2$) between the reduced model ($\mathbf{y}^{(6)}$) and the full model ($\mathbf{y}$) is 0.93, while $R^2$ for the complement model ($\overline{\mathbf{y}^{(6)}}$) and the full model is only 0.06. This demonstrates that the reduced model $\mathbf{y}^{(6)}$, while insufficient to capture all six biomarkers simultaneously, accurately reproduces the single biomarker $A_{90}$. Table 7 lists other specialized reduced

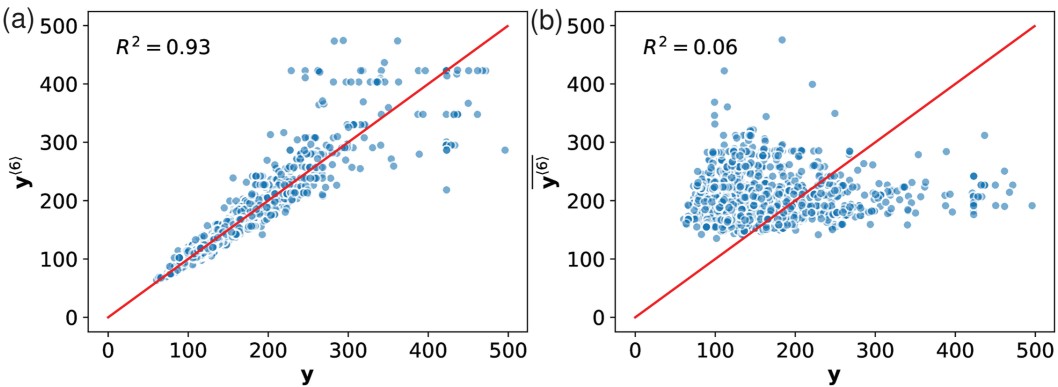

**Fig 9. Scatter plots of $A_{90}$ values obtained from the reduced model $y^{(6)}$ (a) and the reduced model $\overline{y^{(6)}}$ (b) compared to the full model y.**

**Table 7. Minimal sets of parameters for specialised reduced models that capture a specific biomarker with a coefficient of determination value $R^2$ of at least 0.9. Parameters are ranked by their total-order index from large to small. All 15 parameters are required for $A_{30}$ to achieve high accuracy ($R^2 \geq 0.9$) because of their non-negligible high-order interaction effects.**

| AP biomakers | $R^2$ value | Parameters |
|---|---|---|
| $A_{90}$ | 0.93 | $p_{Clb}, p_{Kr}, p_{NaCa}, p_{K1}, p_{CaL}, p_{Ks}$ |
| $A_{30}$ | 1.0 | $p_{Clb}, p_{Kr}, p_{NaCa}, p_{Ks}, p_{CaL}, p_{K1}, p_{NaK}, p_{Cab}, p_{tos}, p_{ClCa}$ $p_{tof}, p_{Nab}, p_{Cap}, p_{Kp}, p_{Na}$ |
| $V_{max}$ | 0.91 | $p_{Clb}, p_{CaL}, p_{K1}, p_{Ks}, p_{Kr}, p_{tos}, p_{tof}, p_{ClCa}, p_{NaCa}, p_{Cab}, p_{NaK}$ |
| $V_{rest}$ | 0.92 | $p_{Clb}, p_{Kr}, p_{K1}, p_{Ks}, p_{NaCa}, p_{CaL}, p_{NaK}, p_{Cab}, p_{tos}, p_{ClCa},$ $p_{Cap}, p_{tof}, p_{Nab}, p_{Na}$ |
| $B$ | 0.91 | $p_{Clb}, p_{Kr}, p_{NaCa}, p_{Ks}, p_{CaL}, p_{K1}, p_{ClCa}, p_{tos}$ |
| $V_{plt}$ | 0.92 | $p_{Clb}, p_{K1}, p_{Kr}, p_{CaL}, p_{NaCa}, p_{tos}, p_{ClCa}, p_{Cab}, p_{NaK}, p_{Ks}$ |

models that capture each of the six biomarkers with $R^2 \geq 0.9$, a threshold often considered sufficient to assess how well one variable replicates another.

The appropriate accuracy of a Shannon model reduction depends on the specific application is intended to investigate. The discrepancy analysis presented in this section provides a framework for selecting a hierarchical model that balances simplicity and accuracy to meet the requirements of the application.

## Conclusion

This study presents a global sensitivity analysis of the Shannon model of rabbit ventricular myocyte electrophysiology, focusing on the influence of ionic current strength parameters on key AP biomarkers. By ranking parameters using Sobol sensitivity indices, we have identified the most influential ionic currents, including $p_{Clb}, p_{Kr}, p_{K1}, p_{CaL}, p_{NaCa}$ and $p_{Ks}$, which dominate the model's variability.

Our analysis demonstrates the utility of a global sensitivity approach in capturing both direct effects and interaction effects among parameters, providing a deeper understanding of their role in shaping AP dynamics. For instance, $p_{Ks}$, which exhibited low first-order indices but high total-order indices, was found to contribute significantly through interaction effects, highlighting the importance of considering higher-order sensitivity metrics.

The hierarchical model framework constructed in this study offers a practical approach to reducing model complexity while maintaining accuracy. Models that exclude less influential parameters ($M \geq 8$) were shown to adequately approximate the full model, as evidenced by monotonic reductions in mean relative error and increases in $R^2$. Additionally, specialized reduced models targeting individual biomarkers were developed, achieving high accuracy ($R^2 \geq 0.9$) while requiring fewer parameters. The findings provide a foundation for tailoring the Shannon model to other specific applications, balancing simplicity with accuracy. The insights gained into parameter importance and interaction effects can guide future investigations, including the development of reduced models for multi-scale simulations and the exploration of electrophysiological variability across different cell types and conditions. As an example, in our prior work on inter-cell variability of rabbit ventricular electrophysiology [3,57] parameter choices were made by educated guesses. The results of the current analysis complement these works and provide justification for their parameter choices.

A notable limitation of this study is its focus on sensitivity analysis at a fixed stimulation rate of 2 Hz, without systematically exploring AP dynamics at other pacing rates. This narrow scope may limit the generalizability of the findings across diverse physiological and pathological conditions where cellular pacing rates vary. Future studies should extend this analysis to include a broader range of stimulation frequencies to enhance applicability. Additionally, the choice and number of model parameters and biomarkers warrant further investigation. While this study focused on a subset of parameters, the Shannon model includes nearly 200 parameters, and incorporating biomarkers such as restitution properties or AP duration rate adaptation could provide a more comprehensive perspective. However, currently there are no experimental studies that assess the intracellular calcium, restitution, or other biomarkers in addition to AP biomarkers. Expanding the sensitivity analysis to alternative AP models, such as the [17] model for rabbit ventricular myocytes, would also enhance the robustness of the findings. Our sensitivity analysis reflects the behavior of the Shannon model as currently formulated. However, the assumption of constant intracellular chloride concentration in the model is a simplification that may exaggerate the influence of $p_{Clb}$. Future extensions incorporating dynamic chloride ion homeostasis will be crucial to reassessing the physiological relevance of this finding. Nonetheless, the result highlights a potentially unappreciated role of chloride currents in AP dynamics, meriting further experimental and modelling investigation.

From a technical standpoint, alternative sensitivity analysis methods offer potential avenues for improving accuracy and efficiency. Comparative studies could evaluate whether these methods yield better insights or computational benefits. The parameter space dimension reduction proposed in this study provides valuable guidance for selecting fitting parameters in sample-specific electrophysiological modeling, enabling the study of specific cellular states and predictions of cellular behavior under varied external conditions.

## Author contributions

**Conceptualization:** Zhechao Yang, Radostin D. Simitev.

**Data curation:** Zhechao Yang.

**Formal analysis:** Zhechao Yang, Godfrey L. Smith, Radostin D. Simitev.

**Funding acquisition:** Radostin D. Simitev.

**Investigation:** Zhechao Yang, Hao Gao.

**Methodology:** Hao Gao, Radostin D. Simitev.

**Project administration:** Radostin D. Simitev.

**Resources:** Radostin D. Simitev.

**Software:** Zhechao Yang.

**Supervision:** Hao Gao, Radostin D. Simitev.

**Writing – original draft:** Zhechao Yang, Radostin D. Simitev.

**Writing – review & editing:** Hao Gao, Godfrey L. Smith, Radostin D. Simitev.

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
