## [Decision Letter · Decision Letter 0]

25 Apr 2025

PONE-D-25-13282Dominant ionic currents in rabbit ventricular action potential dynamicsPLOS ONE

Dear Dr. Simitev,

Thank you for submitting your manuscript to PLOS ONE. After careful consideration, we feel that it has merit but does not fully meet PLOS ONE’s publication criteria as it currently stands. Therefore, we invite you to submit a revised version of the manuscript that addresses the points raised during the review process.

We look forward to receiving your revised manuscript.

Kind regards,

Pan Li, PhD

Academic Editor

PLOS ONE

2. Please update your submission to use the PLOS LaTeX template. The template and more information on our requirements for LaTeX submissions can be found at http://journals.plos.org/plosone/s/latex

“This work was supported by the UK Engineering and Physical Sciences Research Council

[grant numbers EP/S030875/1 and EP/T017899/1].”

Reviewers' comments:

Reviewer's Responses to Questions

**Comments to the Author**

1. Is the manuscript technically sound, and do the data support the conclusions?

Reviewer #1: Yes

Reviewer #2: Partly

Reviewer #3: Yes

2. Has the statistical analysis been performed appropriately and rigorously? 

Reviewer #1: Yes

Reviewer #2: I Don't Know

Reviewer #3: Yes

3. Have the authors made all data underlying the findings in their manuscript fully available?

Reviewer #1: Yes

Reviewer #2: Yes

Reviewer #3: Yes

4. Is the manuscript presented in an intelligible fashion and written in standard English?

Reviewer #1: Yes

Reviewer #2: Yes

Reviewer #3: Yes

5. Review Comments to the Author

Reviewer #1: This paper presents a detailed sensitivity analysis of a rabbit model of ventricular cells.

Similar investigations have been published in the past, but I liked how this study was conducted.

The rationale for studying this specific rabbit model was convincing, considering the experimental variability observed in a cited paper. The methods are sound and sufficiently self-contained, and the results are well presented.

I have a few questions/comments and some minor points.

QUESTIONS

- The weights w_k were set to 1, but the biomarkers vary a lot in magnitude (B in particular).

Would the results be different if B was divided by 1000? I think that based on (5a) it should not be different, but I felt that mentioning this invariance near line 193 would be reassuring.

- Besides the effect of pacing frequency, an issue with single cell models is the effect of stimulus strength.

It would be useful to (1) document the stimulus current applied as compared to the stimulus threshold (minimum to elicit an AP or a sequence of APs), and (2) vary the stimulus between 1x and 2x the threshold and measure the effect on the biomarkers, just to see how it compares to the amplitude of the effect of the p_i.

- Why did you not use p_Na values below 1? Sodium blockers are common antiarrhythmic drugs. Is it in any way related to the choice of stimulus current? The comment at line 390 makes me wonder. Some comments in the discussion/conclusion would be helpful.

- About the Clb current, I'm wondering how long (number of beats) it takes to have an effect. Is it a slow drift that takes minutes or more?

MINOR POINTS

- Table 3: I think that what you call resting potential is generally called maximal diastolic potential because it's measured during pacing. This could be mentioned in the table.

- Eq.(5b): Why is the first summation over all i != j and not only the values of j such that i < j ?

Does S_ii even exist?

- line 189: Should it be "parameters p_i" considering (5c)?

- line 191: "the the"

- lines 224-226: A reference here would be useful.

- section 3.2: Maybe I missed it, but is the random distribution of parameters p_i uniform over the range given in Table 4?

- Table 4: Why did you not use p_Na values below 1? Sodium blockers are common antiarrhythmic drugs.

- Fig. 3: "of the the background..."

- ref 13: incomplete name of the second author

Reviewer #2: This work performs a global sensitivity analysis of the Shannon model and proposes six key parameters to capture individual parameters. The work is interesting but I have some concerns. From my point of view, the manuscript would benefit from some improvements in the methodology and about what is said about the earlier literature.

Major:

1.-In this study, the last action potential of a train of 1000 AP was used for the sensitivity analysis. The authors of Romero et al. (2011) state in the paper that a train of 1000 pulses were enough for reaching the steady state in the simulations performed in that study except for the cases of 50% INaK block and 70% INaCa block, where 2000 pulses were required. In that study, the simulated severe blocks were 100% block of IKr, IKs and Ito, 70% block of INaCa and IK1, 60% block for ICaL and 50% block for INaK. To the best of my knowledge, changes in ionic currents that markedly modify [Ca2+]i and [Na+]i take longer to reach the steady state than changes in ionic currents that do not. As in the present study severe changes in the ionic currents are introduced it is very likely that a train of 1000 pulses is not enough to reach the steady-state in many simulations, especially for changes bigger than +-30% in ionic currents that exert an important influence in the abovementioned concentrations.

2.- The result that IClb is the most influential current is surprising as it is a background current. The fact that the amplitude of this current in the Shannon models is not small can explain its importance. In this model, the intracellular concentration of Cl is constant, so the alteration of this current does not influence this concentration in the simulations, but it the reality this concentration could be significantly altered when altering IClb so much. It could hamper the relevance of this result.

3.- This work does not consider biomarkers of the intracellular calcium concentration. It is true that most experimental studies do not “assess the intracellular calcium, restitution, or other biomarkers in addition to AP biomarkers”. However, there are experimental works that study them and have shown the importance of the consequences of altered the calcium transients. Therefore, many studies using populations of models use biomarkers of intracellular calcium concentration to calibrate them, (https://doi.org/10.1073/pnas.2104019118; https://doi.org/10.1016/j.cmpb.2023.107860; 10.1016/j.yjmcc.2015.09.003. Epub 2015 Sep 16). This work would benefit from considering biomarkers of the intracellular calcium concentration. At least, they should be evaluated or shown within the parameter range of normal response.

4.- Equation 1 includes the coeficients “pi” that multiply “Ii”. However, the following sentences do not state what it represents, unlike Ii, or Istim. It should be included. I guess “pi” corresponds to the scaling factors for the currents flowing across the membrane or between internal compartments. Are the coefficients “pi” also included when the concentrations in the different compartments of the cell are calculated?

Minor:

1. Panel D of Fig. 1 is missing.

2. Fig 5: What does Vplt mean?

Reviewer #3: The manuscript by Yang et al. presents the results of a sensitivity analysis of a mathematical model of cardiac action potentials. Sensitivity analyses of such models have been done extensively, albeit most earlier studies have employed local analyses, whereas the present study utilizes a global analysis. What’s more novel is that the authors utilize the results of the analysis to determine how much of the model variability can be accounted for when allowing only a subset of model parameters (the ones the model output is most sensitive to) to vary.

Major comments:

1) The authors identify the background chloride current (ICl,b) as the current that the AP is most sensitive to. Please discuss this result in a broader context. Do the authors think that this carries over to real cells? Is it perhaps suggesting that this current is too large in the baseline model? A large sensitivity to ICl,b has also been reported for the Grandi human atrial AP model (Krogh-Madsen and Christini 10.1063/1.4999475, Grandi et al. 10.1161/CIRCRESAHA.111.253955), which is based on the Shannon model.

2) It seems that the baseline model is close to bifurcation points for both increasing pNaK (1.07 scaling) and pK1 (1.01 scaling). Is this true in general or would a larger stimulus amplitude move the bifurcation points further away from the baseline values? Based on a general robustness argument, one wouldn’t expect the biological cell to operate so close to bifurcations. Does the proximity of the baseline model to bifurcation points suggest that a modification to the baseline model would improve the model?

3) In figure 7, why are scalings set to 0.8 instead of 1? Doesn’t that cause confounding systematic changes in model output (e.g., the APD differences in panel C)?

Minor comments:

1) Parameters are scaled between 10^-4 to 10 times their baseline values. The authors should justify this range: why are such small values necessary and why the asymmetric scaling?

2) The authors should consider including reference to Parikh et al. (DOI 10.3389/fphar.2019.01054) for a Sobol sensitivity analysis of a cardiac model.

6. PLOS authors have the option to publish the peer review history of their article (what does this mean?). If published, this will include your full peer review and any attached files.

Reviewer #1: No

Reviewer #2: **Yes: **Lucía Romero

Reviewer #3: No

---

## [Author Response · Author response to Decision Letter 1]

3 Jun 2025

Note:

Detailed point-by-point responses to all Reviewers' comments are provided in separate PDF files

* Response_to_Reviewer_1.pdf;

* Response_to_Reviewer_2.pdf;

* Response_to_Reviewer_3.pdf

These are appended at the end of this composite PDF file.

Dear Dr. Li,

Thank you for forwarding the comments from the reviewers and for the opportunity to revise our manuscript.

We have revised the manuscript and addressed all points raised by the three reviewers.

Detailed point-by-point responses to all Reviewers' comments are provided in separate PDF files

* Response_to_Reviewer_1.pdf;

* Response_to_Reviewer_2.pdf;

* Response_to_Reviewer_3.pdf

appended further below.

The major changes are summarized below:

* One new figure (Figure 1) has been added to demonstrate steady-state at the number

of pre-pacing beats considered.

* Figures 2 and 5 have been substantially to include additional new data.

* A new section 3.4 has been added to extend the analysis to intracellular calcium biomarkers.

* Three additional references have been included to reflect relevant recent literature.

These additions required a significant number of new simulations to be conducted. In particular, we repeated the full analysis for two additional stimulus amplitudes and performed a further set of simulations for calcium biomarker measurements. The main conclusions of the original manuscript remain unchanged.

All changes in the revised manuscript are marked in red.

We hope the revisions meet the expectations of the reviewers and the editorial team, and we look forward to your response in due course.

Sincerely,

The Authors

---

## [Decision Letter · Decision Letter 1]

18 Jun 2025

PONE-D-25-13282R1Dominant ionic currents in rabbit ventricular action potential dynamicsPLOS ONE

Dear Dr. Simitev,

Thank you for submitting your manuscript to PLOS ONE. After careful consideration, we feel that it has merit but does not fully meet PLOS ONE’s publication criteria as it currently stands. Therefore, we invite you to submit a revised version of the manuscript that addresses the points raised during the review process.

We look forward to receiving your revised manuscript.

Kind regards,

Pan Li, PhD

Academic Editor

PLOS ONE

Journal Requirements:

Reviewers' comments:

Reviewer's Responses to Questions

**Comments to the Author**

1. If the authors have adequately addressed your comments raised in a previous round of review and you feel that this manuscript is now acceptable for publication, you may indicate that here to bypass the “Comments to the Author” section, enter your conflict of interest statement in the “Confidential to Editor” section, and submit your "Accept" recommendation.

Reviewer #1: All comments have been addressed

Reviewer #2: (No Response)

Reviewer #3: All comments have been addressed

2. Is the manuscript technically sound, and do the data support the conclusions?

Reviewer #1: Yes

Reviewer #2: Partly

Reviewer #3: Yes

3. Has the statistical analysis been performed appropriately and rigorously? 

Reviewer #1: Yes

Reviewer #2: I Don't Know

Reviewer #3: Yes

4. Have the authors made all data underlying the findings in their manuscript fully available?

Reviewer #1: Yes

Reviewer #2: Yes

Reviewer #3: Yes

5. Is the manuscript presented in an intelligible fashion and written in standard English?

Reviewer #1: Yes

Reviewer #2: Yes

Reviewer #3: Yes

6. Review Comments to the Author

Reviewer #1: (No Response)

Reviewer #2: Some of my comments have been correctly addressed but I still have two major concerns.

The first one is about the computation of the ionic concentrations. Based on the manuscript and the answers of the authors, I understand that pi factors are considered in Equation 1, which is used to calculate the membrane potential once the currents are known, but pi factors are not considered when the derivative of the ionic concentrations are calculated taking into account the corresponding ionic currents. Equations 2- 6 of Shannon et al. 2004 (DOI: 0006-3495/04/11/3351/21) describe the derivative of ionic concentrations. The derivative of the ionic concentrations depend on the magnitude of the currents, which are modulated by pi. Therefore, pi must scale their respective currents in these equations as well.

The second one is about the number of pulses needed to reach the steady-state. New Figure 1 depicts the evolution of the biomarkers as a function of the number of beats. They seem stable at pulse 1000. However, all the biomarkers appear in the same graph using a logarithmic scale ranging from 1 to 20000, which makes difficult to see small differences. Moreover, the magnitude of [Ca2+]i is not included. I suspect that not considering the scaling factors of the currents (pi) when computing the ionic concentrations favours reaching the steady state at shorter times.

Reviewer #3: The authors have addressed my concerns and questions in the revised manuscript. I have no further comments or questions

7. PLOS authors have the option to publish the peer review history of their article (what does this mean?). If published, this will include your full peer review and any attached files.

Reviewer #1: No

Reviewer #2: No

Reviewer #3: No

---

## [Author Response · Author response to Decision Letter 2]

24 Jun 2025

Specific minor comments were received form Reviewer 2 only. Our response is included in a separate PDF file "Response_to_Reviewer_2_round_2.pdf". Changes are marked in red in "Tracked_Changes_Yang_etal_20250620_PONE-D-25-13282-R2.pdf".

---

## [Decision Letter · Decision Letter 2]

29 Jun 2025

Dominant ionic currents in rabbit ventricular action potential dynamics

PONE-D-25-13282R2

Dear Dr. Simitev,

We’re pleased to inform you that your manuscript has been judged scientifically suitable for publication and will be formally accepted for publication once it meets all outstanding technical requirements.

Kind regards,

Pan Li, PhD

Academic Editor

PLOS ONE

Additional Editor Comments (optional):

Reviewers' comments:

Reviewer's Responses to Questions

**Comments to the Author**

1. If the authors have adequately addressed your comments raised in a previous round of review and you feel that this manuscript is now acceptable for publication, you may indicate that here to bypass the “Comments to the Author” section, enter your conflict of interest statement in the “Confidential to Editor” section, and submit your "Accept" recommendation.

Reviewer #2: All comments have been addressed

2. Is the manuscript technically sound, and do the data support the conclusions?

Reviewer #2: Yes

3. Has the statistical analysis been performed appropriately and rigorously? 

Reviewer #2: Yes

4. Have the authors made all data underlying the findings in their manuscript fully available?

Reviewer #2: Yes

5. Is the manuscript presented in an intelligible fashion and written in standard English?

Reviewer #2: Yes

6. Review Comments to the Author

Reviewer #2: (No Response)

7. PLOS authors have the option to publish the peer review history of their article (what does this mean?). If published, this will include your full peer review and any attached files.

Reviewer #2: **Yes: **Lucia Romero

---

## [Editor Report · Acceptance letter]

PONE-D-25-13282R2

PLOS ONE

Dear Dr. Simitev,

I'm pleased to inform you that your manuscript has been deemed suitable for publication in PLOS ONE. Congratulations! Your manuscript is now being handed over to our production team.

Kind regards,

on behalf of

Dr. Pan Li

Academic Editor

PLOS ONE